# Exploring the Therapeutic Effects of *Atractylodes macrocephala* Koidz against Human Gastric Cancer

**DOI:** 10.3390/nu16070965

**Published:** 2024-03-27

**Authors:** Na-Ri Choi, Woo-Gyun Choi, Anlin Zhu, Joon Park, Yun-Tai Kim, Jaewoo Hong, Byung-Joo Kim

**Affiliations:** 1Department of Longevity and Biofunctional Medicine, Pusan National University School of Korean Medicine, Yangsan 50612, Republic of Korea; nariring@gmail.com (N.-R.C.); ak0510@hanmail.net (W.-G.C.); 2Department of Physiology, Daegu Catholic University School of Medicine, Daegu 42472, Republic of Korea; zhuanlin@cu.ac.kr; 3Division of Food Functionality, Korea Food Research Institute, Wanju-gun 55365, Republic of Korea; biosciencepark@gmail.com (J.P.); ytkim@kfri.re.kr (Y.-T.K.); 4Department of Food Biotechnology, Korea University of Science & Technology, Daejeon 34113, Republic of Korea

**Keywords:** *Atractylodes macrocephala* Koidz, traditional herbal medicine, AGS human gastric adenocarcinoma cell, anti-cancer, xenograft

## Abstract

*Atractylodes macrocephala* Koidz (AMK) is a traditional herbal medicine used for thousands of years in East Asia to improve a variety of illnesses and conditions, including cancers. This study explored the effect of AMK extract on apoptosis and tumor-grafted mice using AGS human gastric adenocarcinoma cells. We investigated the compounds, target genes, and associated diseases of AMK using the Traditional Chinese Medical Systems Pharmacy (TCMSP) database platform. Cell viability assay, cell cycle and mitochondrial depolarization analysis, caspase activity assay, reactive oxygen species (ROS) assay, and wound healing and spheroid formation assay were used to investigate the anti-cancer effects of AMK extract on AGS cells. Also, in vivo studies were conducted using subcutaneous xenografts. AMK extract reduced the viability of AGS cells and increased the sub-G1 cell fraction and the mitochondrial membrane potential. Also, AMK extract increased the production of ROS. AMK extract induced the increased caspase activities and modulated the mitogen-activated protein kinases (MAPK). In addition, AMK extract effectively inhibited AGS cell migration and led to a notable reduction in the growth of AGS spheroids. Moreover, AMK extract hindered the growth of AGS xenograft tumors in NSG mice. Our results suggest that AMK has anti-cancer effects by promoting cell cycle arrest and inhibiting the proliferation of AGS cancer cells and a xenograft model through apoptosis. This study could provide a novel approach to treat gastric cancer.

## 1. Introduction

Gastric cancer ranks as the fifth most common cancer in world [1,2]. Without early detection, it is challenging to treat cancer, and patients and their families experience excruciating pain and severe financial losses, resulting in significant social suffering [3]. As the emphasis shifts from mere life extension to improving the quality of life, including overall health and well-being, there is a growing recognition of the heightened significance of cancer prevention and early intervention [4]. It is very necessary to develop new drugs that can increase the survival rate of gastric cancer patients and relieve pain.

The rhizomes of *Atractylodes macrocephala* Koidz (AMK; Baegchul in Korea, Báizhú in China) have been used as a traditional medicine for a long time in Korean, Japanese, and Chinese medicine. It is used to treat a variety of illnesses and conditions, such as cancer, Alzheimer’s disease, and other diseases, as well as gastrointestinal disorders, spleen dysfunction, appetite loss, abdominal distension, diarrhea, dizziness, and heart palpitations [5]. Apoptosis is key to maintaining the balance between cell division and death. Uncontrolled cell division caused by errors in apoptosis can result in diseases like cancer [6]. Therefore, apoptosis has emerged as a crucial component of cancer therapy, and the process underlying cancer cell development has emerged as a prominent area of focus in tumor therapy [7]. However, research on the anti-cancer effects of AMK on gastric cancer cells remains limited. This study examined the mechanisms behind these effects to understand how AMK extract affects AGS human gastric cancer cells.

For this, we first used a network-based systems pharmacology approach to assess the effect of AMK on gastric cancer and to understand this relationship better. In our study, we leveraged the Traditional Chinese Medicine Systems Pharmacology (TCMSP) database and its analytical platform, which offers extensive data on herbs, their constituent compounds, molecular target interactions, disease associations, and valuable insights into drug absorption, distribution, metabolism, and excretion (ADME) properties. This study provided the relationship between AMK and gastric cancer using network-based systems pharmacology, and the information obtained may be utilized to develop new treatment or prevention strategies.

## 2. Materials and Methods

### 2.1. Network Pharmacological Analysis of AMK

#### 2.1.1. Identification and Screening of Active Compounds of AMK

We identified potential active compounds in AMK using TCMSP (https://tcmsp-e.com/tcmsp.php, accessed on 19 April 2023). The herb name was searched by entering ‘*Atractylodes macrocephala* Koidz’. Specific ADME parameters were used to determine the active compounds. Active compounds meeting the criteria were chosen for further analysis (OB ≥ 30%, DL ≥ 0.10, and Caco-2 ≥ −0.4).

#### 2.1.2. Analysis of Targets

We obtained the target information for the compound by querying TCMSP, and the target proteins were subsequently associated with their official gene names through the UniProt database (https://www.uniprot.org/uniprot, accessed on 19 April 2023) [8].

### 2.2. In Vivo and In Vitro Experiments

#### 2.2.1. Sample Preparation

The AMK, sourced from the Korea Plant Extract Bank (Ochang, Chungbuk, Republic of Korea) was obtained by subjecting AMK to a 3-day extraction process using 95% ethanol. The extract was then filtered and evaporated to dryness. The final AMK product was reconstituted in DMSO and diluted to the desired concentrations. From J.T. Baker (Phillipsburg, NJ, USA), high performance liquid chromatography (HPLC)-grade acetonitrile (ACN) and water were purchased. Eudesma-4(15),7(11)-dien-8-one (EDO) (Chemfaces, Wuhan, China), and Atractylenolide III (ATO III) (Sigma-Aldrich, St. Louis, MO, USA) were used.

#### 2.2.2. HPLC Analysis

Chromatographic analysis was conducted using a JASCO HPLC system comprising an autosampler, a column oven, and a UV detector. The UV detector was set at a wavelength of 236 nm. For analysis, a 10 μL volume of either standard or AML sample solutions was directly injected into a Symmetry 300TM C18 column (4.6 × 250 mm × 5μm). The mobile phase consisted of a mixture of (A) ACN and (B) HPLC-grade water (*v*/*v*). The gradient elution conditions were 50–65% A for the first 0–35 min, followed by 65–100% A for the next 35–50 min. A re-equilibration period of 5 min was allowed between sample injections.

#### 2.2.3. Cell Culture and Cell Viability Assay

The AGS cell line was purchased from the Korean Cellular Bank. The cells were cultivated with an RPMI-1640 medium containing 10% serum (Cytiva, Marlborough, MA, USA). Cell viability was measured using 3-(4,5-dimethylthiazole-2-day)-2,5-diphenyltetrazolium bromide (MTT) assay after treatment with AMK. Cell count kit-8 (CCK-8) was used after AMK extract treatment for 24 h.

#### 2.2.4. Cell Cycle Analysis

AGS cells were treated with AMK extract for 24 h, then collected and fixed in cold ethanol. After centrifugation for 5 min, the cell pellets were reconstituted. Cell pellets were stained with a solution mixed with RNase and propidium iodide (PI) for 40 min. Subsequently, a fluorescence-activated cell sorter (FACS CANTO II; Becton Dickinson (BD) Biosciences, San Jose, CA, USA) was used to assess the samples.

#### 2.2.5. Intracellular Reactive Oxygen Species (ROS) Levels

Similar to previous studies, we used flow cytometry to quantify the intracellular ROS production using DCFH-DA [9]. The cells were treated with the appropriate drugs for two h and then incubated with DCFH-DA. After collection, the cells were immediately examined using a flow cytometer after washing with PBS. The fluorescence levels of 10,000 cells were also examined using BD FACSDIVA software (Version 9.0).

#### 2.2.6. Mitochondrial Depolarization Assay

AGS cells treated with the designated drugs were incubated with 100 nM tetramethylrhodamine methyl ester (TMRM; Sigma-Aldrich, St. Louis, MO, USA) for 30 min. Subsequently, the fluorescence intensities were checked using FACS CANTO II.

#### 2.2.7. Caspase Assay

Caspase assays were conducted using Caspase-3 and -9 detection kits (BioVision Inc., Milpitas, CA, USA). The experiments were conducted according to the instructions of the kits. The absorbance of each sample was checked at 405 nm.

#### 2.2.8. Western Blot Analysis

Cell lysates were prepared in RIPA buffer, and the total protein content was quantified using the Bradford method. Protein from the samples was separated by SDS-PAGE, with gel percentages of 8%, 10%, and 12%, and subsequently probed with specific antibodies. The relative intensities of protein bands were analyzed using an ImageQuant™ LAS 4000 system (GE HealthCare Bio-Sciences AB, Hercules, Uppsala, Sweden). Results are representative of a minimum of three independent experiments.

#### 2.2.9. Wound Healing Assay

Cell migration capacity was assessed using a wound-healing assay performed on AGS cells. AGS cells were initially placed in a culture insert (ibidi GmbH, Martinsried, Germany). Each well was filled with 70 µL of a cell suspension solution, diluted to a concentration of 3.0 × 10^5^ cells/mL, and incubated for 24 h. Images were captured from selected areas in each well using a Nikon Corporation camera (Tokyo, Japan), before and after 24 h of AMK treatment. The degree of migration relative to the control group was used with Image J version 1.52a (National Institutes of Health, Bethesda, MD, USA).

#### 2.2.10. Spheroid Formation Assay

AGS cells were cultured in Corning^®^ (Corning, NY, USA) spheroid microplates at a density of 5 × 10^3^ cells/well and incubated for three days at 37 °C. Images of the cultivated spheroids were captured three days after seeding to investigate AGS spheroid formation. These spheroids were pre-treated with or without 10 µM z-VAD (pan-caspase inhibitor) before being treated with the stated doses of AMK extract for six days. The medium was changed every two days. The spheroid size was quantified using NIS-Elements BR software (Version 4.3.; Canon, Ota, Japan).

#### 2.2.11. Preparation of AGS-iRFP Cells

iRFP-C1 (Addgene, Watertown, MA, USA) was transfected into AGS cells using PEI to generate AGS-iRFP cells for in vivo imaging. The cells were selected using the G418. Briefly, the cells were 70% confluent in a 3 cm plate. Then, 0.5 μg of DNA was mixed with 2.5 μg of PEI for 10 min and added to the culture medium. The medium was replaced after a brief wash with PBS. The transfected cells were chosen with 0.5 mg/mL G418 for 7 days for further experiments.

#### 2.2.12. Animal Experiment

A total of 5 × 10^6^ AGS-iRFP cells in 100 μL PBS were inoculated subcutaneously in the flank of 5-week-old female NSG mice to establish the gastric adenocarcinoma model. Tumor growth was monitored using a VISQUE InVivo Smart-LF instrument (South Korea). Mice were randomly divided into groups receiving 0, 10, or 50 mg/kg AMK (*n* = 5/group in all efficacy studies), and dosing was initiated on day 0 after tumor cell injection (5 times/week). Tumor images and body weights were measured weekly.

#### 2.2.13. Statistical Analysis

The results were presented as means ± SEMs, and Tukey’s multiple comparison test was used for the analysis of variance (ANOVA) using GraphPad Prism 6. Statistical significance was set at *p* values < 0.05.

## 3. Results

### 3.1. Network-Based Pharmacological Analysis

#### 3.1.1. Eighteen Active Compounds Meet the ADME Parameter Criteria

55 potentially active compounds were identified in AMK using the TCMSP database (Appendix A), and 18 compounds meeting the ADME criteria were considered the active compounds (Appendix A). In addition, among 55 potentially active compounds, 50 compounds contained target information. It has been shown that these compounds and 260 targets interacted through a combination of 774 compounds (Appendix A). GLY was linked to the most significant number of target genes (173 genes), followed by LPG (72 genes), ASI (50 genes), D-Serin (40 genes), Glutamine (30 genes), L-Valine (28 genes), PHA (25 genes), DTY (21 genes), Prolinum (20 genes), atactylone (18 genes), palmitic acid (17 genes), and 3β-acetylone (16 genes).

#### 3.1.2. Identification of 37 Compounds Related to Gastrointestinal (GI) Cancer in AMK

We investigated compound–target–disease relationships using the TCMSP database. Our analysis identified 37 compounds associated with GI cancer (Table 1). Notably, atractylenolide I, atractylenolide III, 3β-acetoxyatractylone, (5E,9Z)-3,6,10-trimethyl-4,7,8,11-tetrahydrocyclodeca[b]furan, atra-ctyl-one, 14-acetyl-12-senecioyl-2E,8Z,10E-atractylentriol, 8β-ethoxy atractylenolide III, Akridin, juniper camphor, selina-4(14),7(11)-dien-8-one, and α-Longipinen were identified as active compounds associated with GI cancer (Figure 1).

#### 3.1.3. Six Genes Are Included in Both Gastric-Cancer-Related Genes and AMK Target Genes

To establish a link between AMK and gastric cancer among GI cancer, we initially examined genetic information pertaining to gastric cancer using Cytoscape. We identified 100 genes associated with gastric cancer by applying a score threshold of 0.40, with 100 proteins (Table 2). Subsequently, we constructed a network encompassing GC-related genes and AMK target genes (Figure 2), revealing six genes common to both gene sets. The genes targeted by AMK in the context of gastric cancer include apoptosis regulator Bcl-2 (BCL2), catenin beta 1 (CTNNB1), B-cell stimulatory factor 2 (IL6), mutated in multiple advanced cancers 1 (PTEN), proto-oncogene (SRC), and tumor necrosis factor ligand member 2 (TNF).

#### 3.1.4. The Network of AMK Compounds and Gastric-Cancer-Related Genes

The results present a network that reveals associations between the active compounds of AMK and the target genes associated with gastric cancer (Figure 3). TNF was shown to have the strongest association with gastric cancer. In summary, palmitic acid, atractylenolide I, GLY, beta-caryophyllene, D-Serine, LPG, alpha-humulene, and atractylenolide III were identified as active substances that specifically target genes related to gastric cancer. These findings suggest that these compounds are potential candidates for drug development.

### 3.2. In Vitro Experiments

#### 3.2.1. Quantification of Major Components in AMK by HPLC-UV

Previous studies highlighted EDO and ATO III as prominent constituents of AMK [10]. A chromatographic analysis was performed to confirm the major components. HPLC results showed that the contents of EDO and ATO III were found to be 18.77 ± 0.43 mg/g and 13.83 ± 0.52 mg/g in AMK, respectively (Appendix A).

#### 3.2.2. Effects of AMK Extract on Cytotoxicity in AGS Cells

We evaluated the cytotoxicity of AMK extract in AGS cells using a series of experiments. AGS cells were exposed to varying concentrations of AMK extract, followed by assessment using the MTT assay. These results clearly demonstrated the concentration-dependent inhibition of cell proliferation following AMK extract treatment. Specifically, AMK extract for 24 h led to a significant reduction in cell viability to 76.02% at 50 µg/mL, 59.20% at 100 µg/mL, 29.28% at 150 µg/mL, and 6.86% at 200 µg/mL, respectively (Figure 4A). With an extended incubation time of 48 h, cell viability decreased further to 57.20%, 41.68%, 4.72%, and 3.10%, respectively (Figure 4B). After 72 h, cell viability decreased to 51.43%, 43.72%, 3.47%, and 3.47%, respectively (Figure 4C). To further elucidate the impact of AMK extract on AGS cells in vitro, a CCK8 assay was employed to measure cell viability (Figure 4D). Exposure to different concentrations of AMK extract for 24 h resulted in a reduction in the AGS cell growth. Specifically, treatment with AMK extract decreased cellular viability by 76.05% at 50 μg/mL, 36.55% at 100 μg/mL, 16.65% at 150 μg/mL, and 13.73% at 200 μg/mL.

#### 3.2.3. AMK Extract Induces Cell Cycle G1 Arrest in AGS Cells

We investigated whether the growth reduction in AGS cells induced by AMK extract was linked to apoptosis, a known mechanism associated with many anti-cancer agents. One characteristic of various anti-cancer agents is their interference with cell cycle progression [11]. Flow cytometry was performed to assess the effect of AMK extract on the cell cycle of AGS cells. As anticipated, 24 h treatment with AMK extract led to a significant increase in AGS cells in the sub-G1 phase (Figure 5).

#### 3.2.4. AMK Extract Induces Apoptosis through ROS Generation in AGS Cells

Disruption of cellular redox balance is essential to the pro-apoptotic effect of various anti-cancer drugs [12,13]. Overproduction of ROS and free radicals can inflict substantial harm on lipids, proteins, and DNA, thereby influencing the mechanisms underlying the initiation of autophagy and apoptosis [14,15]. Consequently, we conducted DCF-DA staining to assess the impact of AMK extract on intracellular ROS production and employed the ROS scavenger NAC to investigate its antagonistic effects. Flow cytometry analysis revealed a notable increase in ROS levels upon AMK extract treatment of AGS cells (Figure 6A,B). Furthermore, when AGS cells were treated with 200 µg/mL AMK extract, pre-treatment with NAC significantly suppressed the increased ROS generation.

#### 3.2.5. Alterations in the Mitochondrial Membrane Potential in AGS Cells

In the context of cancer cell research, mitochondrial membrane potential serves as a crucial indicator for evaluating mitochondrial function and health. To ascertain whether the cytotoxic effects of AMK extract are linked to the mitochondrial membrane, we quantified mitochondrial depolarization by analyzing TMRM fluorescence intensity using flow cytometry. As illustrated in Figure 6C,D, our findings revealed a significant increase in mitochondrial depolarization following AMK extract treatment. Notably, AMK extract treatment reduced TMRM-positive fluorescence in AGS cells. Consequently, our results strongly suggest that AMK extract diminishes the mitochondrial membrane potential.

#### 3.2.6. AMK Extract Induces the Activation of Caspases in AGS Cells

Caspases act as central mediators in orchestrating apoptosis by participating in intrinsic and extrinsic apoptotic pathways. These caspase enzymes lead to the cleavage of various substrates, of which poly ADP ribose polymerase (PARP) stands out [16]. Our investigation aimed to determine whether AMK extract triggers the caspase signaling pathway. AMK extract increased the cleavage of pro-Caspase-9, indicating its involvement in the intrinsic apoptotic pathway (Figure 7A). Furthermore, we observed a significant increase in caspase-3 and PARP cleavage (Figure 7B), indicating that AMK extract triggered mitochondria-mediated apoptosis in AGS cells. To further validate the apoptotic mechanism, we employed the Z-VAD, which revealed that Z-VAD treatment rescued the AMK extract-induced inhibition of AGS cell proliferation. Moreover, Z-VAD reduced AMK extract-induced activation of Caspase-3 and Caspase-9 (Figure 7A). These findings suggest that AMK extract induces apoptosis in AGS cells by activating intrinsic apoptotic pathways.

#### 3.2.7. Relationship between AMK-Extract-Induced Apoptosis and Mitochondrial Pathway in AGS Cells

To further elucidate the effect of AMK extract treatment on AGS cells, we investigated the alterations in protein expression related to the mitochondrial pathway. The intrinsic apoptotic pathway, central to governing cellular apoptosis and proliferation, primarily involves the BCl-2 family of proteins [17]. Our study revealed that AMK extract treatment resulted in an augmentation of the Bax, while concurrently reducing the levels of the B-cell lymphoma-2 (BCl-2) and B-cell lymphoma-extra large (BCl-xL) (Figure 8A,B). Furthermore, the Bax/Bcl-2 ratio increased in corresponding increments (Figure 8B). Additionally, survivin’s expression, an apoptosis inhibitor, decreased in response to AMK extract treatment (Figure 8C,D). These cumulative findings substantiate the idea that AMK extract-induced apoptosis in AGS cells is closely linked to the activation of the mitochondrial pathway.

#### 3.2.8. AMK Extract-Mediated Regulation of Mitogen-Activated Protein Kinase (MAPK) in AGS Cells

To explore the impact of AMK extract on the MAPK pathway, we used Western blotting to evaluate the phosphorylation status of key components, including extracellular signal-regulated kinase (ERK), c-Jun N-terminal kinase (JNK), p38, and Serine/Threonine protein kinase (AKT). Following time-dependent treatment with 200 μg/mL AMK extract, we observed a significant increase in p38 and JNK phosphorylation, alongside a noticeable reduction in the phosphorylation levels of the ERK and AKT (Figure 9). These observations indicated that AMK extract induces apoptosis by modulating the MAPK pathway in AGS cells.

#### 3.2.9. Effects of AMK Extract on Migration Ability in AGS Cells

AMK extract effectively inhibits AGS cell migration, an important factor in tumor metastasis. Assessment of migratory capacity was performed using a wound healing assay, measured through the area occupied by AGS cells 24 h after treatment with AMK extract. The results indicated a significant decrease in mobility. (Figure 10A,B)

#### 3.2.10. AMK Extract Suppresses the Growth of AGS Cells in the Spheroid Formation Assay

To further validate the anti-cancer potential of AMK extract in a more physiologically relevant in vitro setting, we established a 3D multicellular tumor spheroid (MTS) model. We assessed the effect of AMK extract on spheroid growth. Following a 6-day culture period of AMK extract treatment, the spheroids were examined using an optical microscope. The results demonstrated that AMK extract treatment led to a notable reduction in the growth of AGS spheroids, as evidenced by a decrease in spheroid volume (Figure 10C,D). To elucidate the mechanism of apoptosis in the 3D spheroid model, we used the Z-VAD. Notably, pre-treatment with Z-VAD reversed the proliferation-inhibiting effects of AMK extract on AGS cells. These findings prove that AMK extract triggers apoptosis in AGS cells by activating the intrinsic apoptotic pathway. Moreover, these results are the same as those identified in the 2D monolayer cultures.

### 3.3. In Vivo Experiments

#### AMK Extract Inhibits Tumor Growth in the Xenograft Mouse Models

We established a mouse model of human gastric adenocarcinoma to demonstrate the anti-tumor activity of AMK. The mice’s body weights in each group were not different (Figure 11A). The tumors started to grow from the sixth week, but tumor growth in the AMK extract treatment group was significantly lower than in the non-treated group (Figure 11B). In addition, the iRFP signal of the tumor cells (Figure 11C) and gross growth of the tumor mass (Figure 11D) were significantly suppressed compared to those in the control group mice. After nine weeks of treatment, the tumors were removed and weighed (Figure 11E,F). The tumor size was significantly reduced by treatment with 50 mg/kg AMK extract (*p* = 0.0345). These observations demonstrate that AMK extract decreases tumorigenesis in a mouse model of gastric adenocarcinoma.

## 4. Discussion

Our research commenced with a keen focus on exploring the potential of natural products and herbal remedies for developing novel pharmaceuticals. Specifically, we turned our attention to the rhizome of AMK, commonly called Baizhu, which has a longstanding history of utilization in traditional medicine in Chinese, Japan, and Korea. It improves stomach health, aids digestion, and is effective for chronic indigestion, enteritis, diarrhea, and appetite enhancement. It has been proven particularly beneficial for individuals whose illnesses cause decreased appetite, fatigue, or hyperhidrosis. It is combined with other medicinal herbs to improve the treatment effect on chronic rheumatism of the joints. Considering its potential as a novel therapeutic agent for various diseases, including cancer, AMK is believed to have a notable therapeutic potential [1,18]. This study showed that AMK extract induces apoptosis in AGS cells.

As a result of network pharmacological analysis, 55 compounds containing 18 active compounds in AMK were confirmed. A total of 50 of the 55 compounds had target information, and 260 targeted genes were collected (Appendix A). Among them, GLY was linked to the most significant number of target genes.

In our study, TNF, IL6, BCL2, PTEN, CTNNB1, and SRC are presented as common target genes between gastric cancer and AMK (Figure 2). We found it was difficult to confirm the relevance of these genes during the revision of this manuscript. However, in previous studies, we have validated the relationship between these six genes and gastric cancer. TNF is known to enhance inflammatory responses in gastric cancer [19]. Among TNFs, TNF-alpha is mainly involved in the early stages of gastric cancer [19]. Therefore, TNF-alpha is the most studied marker in gastric cancer, and many studies are being conducted on the development of gastric cancer and the relationship with other gastric cancer biomarkers [20]. IL-6 is one of the most important cytokines in the tumor microenvironment, and many studies have proved that IL-6 is involved in cancer cell growth and metastasis [21]. In gastric cancer patient blood, IL-6 concentration is higher in stage 4 than in stages 2 and 3, so IL-6 may be related to the malignancy of gastric cancer such as metastasis and cancer severity [22]. BCL2 inhibits the mitochondrial pathway of apoptosis and plays an important role in the progression of gastric cancer, which is closely related to the severity of gastric cancer [23]. It is known that BCL2 expression level is increased as gastric cancer progresses [24]. PTEN regulates various cellular processes, such as the induction of apoptosis [25]. When PTEN activity decreases, it is known that it is associated with various cancers, and gastric cancer is also known to be associated with PTEN inactivation [26]. CTNNB1 genes are present in gastric cancer and are involved in the signaling mechanisms of gastric cancer along with various molecules [27]. Various signaling pathways such as Wnt, Notch, and Ephrin signaling pathways have been needed to reveal the CTNNB1 signaling in cancer [27]. SRC has been involved in the growth and metastasis of cancer cells and is also an important target for gastric cancer treatment [28]. c-SRC was the first oncogene discovered and has increased expression in gastric cancer [29,30]. As such, there are many previous reports that the six genes identified by the network analysis are related to gastric cancer, and further clarification is needed in the future to reveal the underlying mechanisms of AMK.

As shown in Figure 3, AMK compounds targeting gastric-cancer-related genes, palmitic acid, atractylenolide I, GLY, beta-caryophyllene, D-Serine, LPG, alpha-humulene, and atractylenolide III, were identified. In addition, palmitic acid was observed to target PTEN and BLC2. Atractylenolide I targeted TNF and IL6, and GLY targeted CTNNB1. Beta-caryophyllene targeted IL6, D-Serine and LPG targeted SRC, and alpha-humulene and atractylenolide III targeted TNF. These results show the characteristics of multiple compounds/multiple targets of traditional medicines, and with the synergistic effect of multiple compounds/multiple targets of AMK, we could predict the therapeutic effects of AMK on gastric cancer.

Apoptosis is a multifaceted, multistep phenomenon encompassing numerous genes that regulate physiological growth and tissue homeostasis [31,32]. The promotion of apoptosis has been established as an effective strategy in cancer treatment. Apoptosis can be divided into two primary pathways: the intrinsic mitochondrial pathway, which triggers apoptosis through direct interactions with intracellular organelles, and the extrinsic death receptor pathway, which initiates apoptosis through receptor-mediated interactions [33,34,35,36].

We performed flow cytometry with FITC staining to investigate the effect of AMK extract on apoptosis. Flow cytometry, using FITC as a fluorescent probe, is a widely employed and highly sensitive technique for discerning various stages of apoptosis. [37,38]. Our results revealed that AMK extract treatment increased the sub-G1 phase in AGS cells, signifying its inhibitory effect on AGS cell growth, likely through the induction of apoptosis.

A critical aspect of our discussion revolves around the role of the Bcl-2 family of proteins in controlling the mitochondrial pathway, focusing on Bax [39,40,41]. Western blot analysis demonstrated that AMK extract treatment upregulated Bax, downregulated Bcl-2 and Bcl-xL and increased the Bax/Bcl-2 ratio. This indicates that AMK extract induces apoptosis through a mitochondria-dependent pathway by regulating apoptosis-related proteins. We also observed a significant decrease in survivin protein levels following AMK extract treatment. Survivin inhibits the apoptotic pathway and interacts with Caspase-3 and Caspase-7. Downregulation of survivin suggests that AMK extract is a potent inducer of apoptosis in AGS cells [42,43,44]. We further highlighted the pivotal role of the proteolytic system involving caspases, with a specific focus on the capacity of AMK extract to upregulate active Caspase-3 and Caspase-9 forms while downregulating pro-Caspase-3 and pro-Caspase-9. Notably, our findings demonstrated that AMK-extract-induced cell death can be suppressed by Z-VAD, indicating that AMK extract induces apoptosis through caspase and mitochondrial pathways [45,46]. Caspases play a central role in regulating cell death and cleave various targets, including PARP. In particular, cleavage of PARP is a unique feature of apoptosis and cleaved PARP is an indicator of programmed cell death.

ERK undergoes phosphorylation and activation in response to mitogenic signals via the Ras/Raf/MEK signaling cascade. Activated p-ERK plays a pivotal role in initiating the transcription of various associated proteins and participates in cell proliferation, apoptosis, and migration processes. AKT, a crucial downstream effector of PI3K, is activated by phosphorylation. Once activated, AKT phosphorylates multiple downstream targets within the survival and apoptotic pathways, thereby executing pivotal functions [47,48]. Our findings clearly illustrated that AMK treatment reduced p-ERK and p-Akt levels, coinciding with the downregulation of Bcl-2. These results strongly imply that AMK extract can potentially impede cell growth and induce apoptosis by influencing the ERK and Akt signaling pathways. In conclusion, our study provides compelling evidence that AMK has anti-cancer effects, suggesting its potential as a therapeutic option for gastric carcinoma.

We conducted a comprehensive analysis, including both in vitro and in vivo experiments, to assess the effects of AMK extract on various aspects of cancer biology. In the in vitro experiments, we focused on inhibiting AGS cell migration using a wound-healing assay and suppressing spheroid growth using a 3D MTS model. These experiments revealed that AMK extracts effectively hindered AGS cell migration, a critical factor in tumor metastasis. Furthermore, in the 3D spheroid model, AMK treatment significantly reduced spheroid growth, indicating its ability to impede tumor progression. Importantly, using the pan-caspase inhibitor Z-VAD provided mechanistic insights, as pre-treatment with Z-VAD effectively reversed the proliferation-inhibiting effects of AMK extract on AGS cells. This suggests that AMK extract triggers apoptosis in AGS cells by activating the intrinsic apoptotic pathway, which is consistent with our observations in 2D monolayer cultures (Figure 12).

For the in vivo experiments, we established a mouse model of human gastric adenocarcinoma to assess the anti-tumor activity of AMK extract. The results were compelling, as we observed a reduction in cancer growth in the AMK extract treatment group compared with that in the non-treated group. Consistent body weights across groups indicated that AMK extract treatment did not cause significant adverse effects in mice. Suppression of the iRFP signal from tumor cells and gross tumor mass growth further support the efficacy of AMK extract in inhibiting tumorigenesis. Notably, after nine weeks of treatment, the tumors were removed and weighed, revealing a significant reduction in tumor size when treated with 50 mg/kg AMK extract.

## 5. Conclusions

These observations demonstrate that AMK extract has potential as a good treatment agent against gastric adenocarcinoma. Its ability to inhibit cell migration, suppress spheroid growth, and reduce tumor size in a mouse model highlights its multifaceted anti-cancer properties. Therefore, our study suggests that AMK extract holds promise as a novel and multifunctional therapeutic option for gastric adenocarcinoma and provides a foundation for further investigation of its detailed mechanisms and clinical applications.

## Figures and Tables

**Figure 1 nutrients-16-00965-f001:**
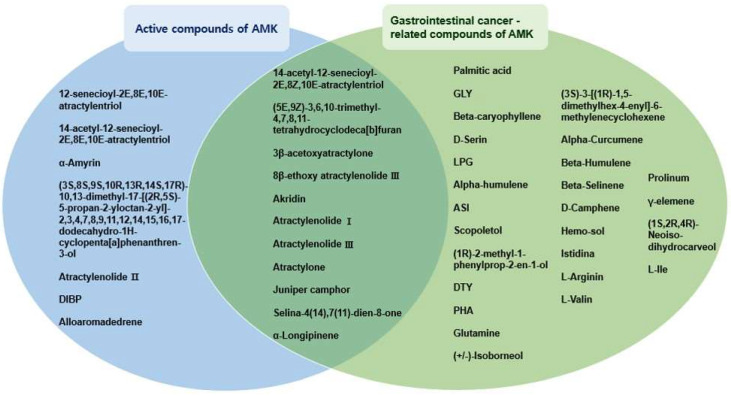
Interactions between active compounds from AMK and GI-cancer-related compounds.

**Figure 2 nutrients-16-00965-f002:**
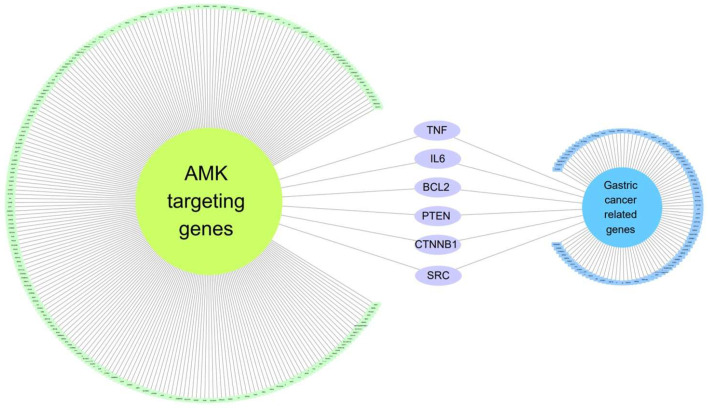
The network of gastric-cancer-related genes and AMK targeting genes. At the network’s core, six genes are found shared between gastric-cancer-related genes and AMK target genes.

**Figure 3 nutrients-16-00965-f003:**
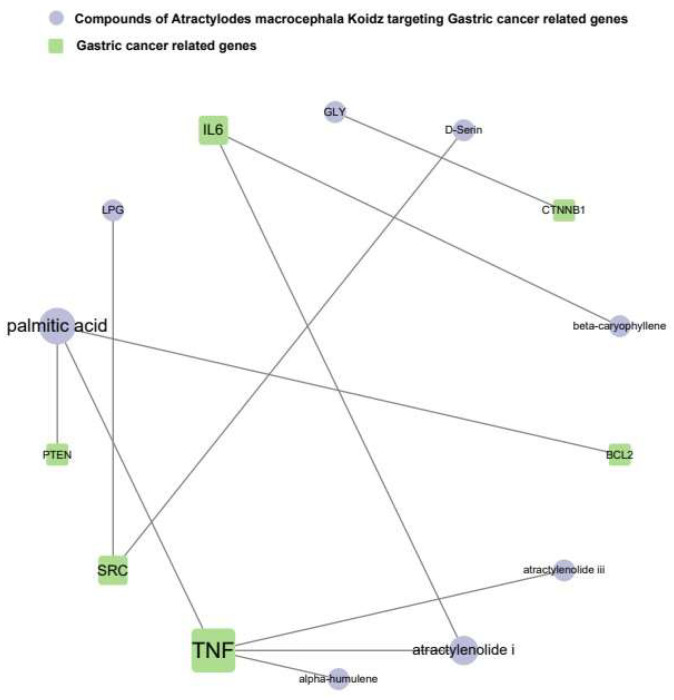
Network of compounds of AMK and gastric-cancer-related genes.

**Figure 4 nutrients-16-00965-f004:**
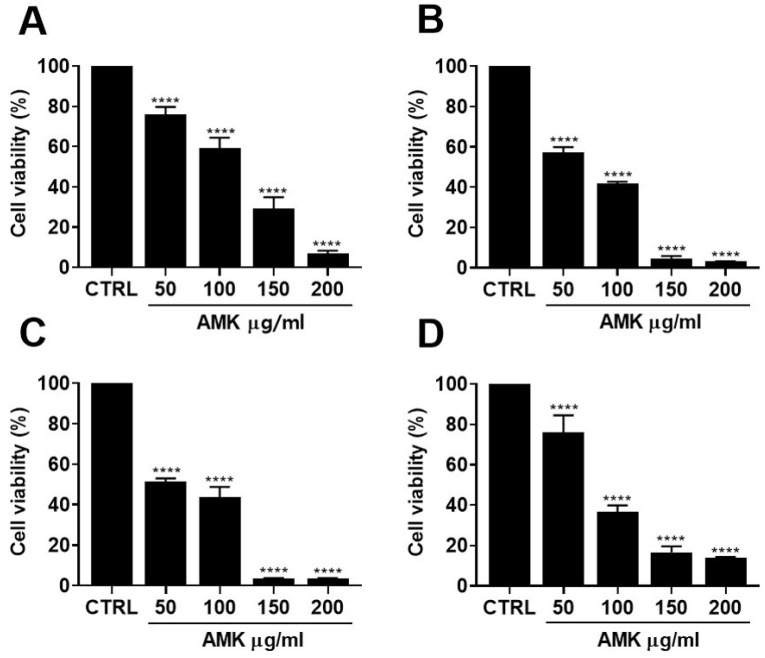
Effects of AMK extract on AGS cell viability. We conducted cell viability assessments at various time points, including (**A**) 24 h, (**B**) 48 h, and (**C**) 72 h after treatment with AMK extract using the MTT assay. Additionally, (**D**) cell viability was determined after 24 h of AMK extract treatment utilizing the cell counting kit-8 assay. The results consistently indicate that AMK extract treatment for 24, 48, and 72 h leads to a notable decrease in cell viability. Results are presented as the means ± standard error. **** *p* < 0.0001 compared to untreated controls. AMK: *Atractylodes macrocephala* Koidz; CTRL: Control.

**Figure 5 nutrients-16-00965-f005:**
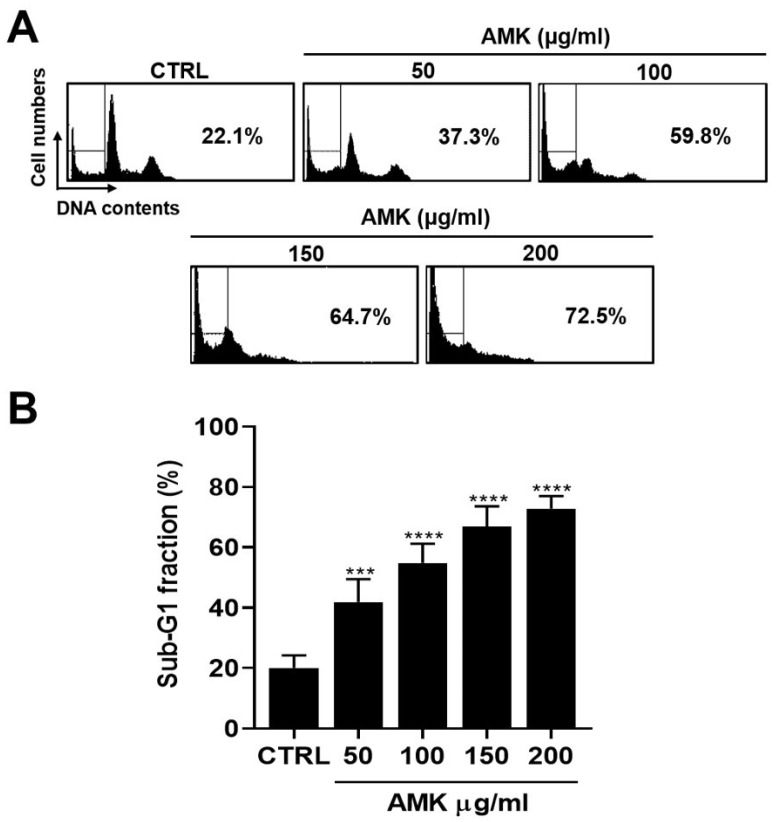
Effects of AMK extract on apoptosis in AGS cells. (**A**) The identification of the Sub-G1 cell fraction via flow cytometry, with (**B**) results presented as percentages. Results are presented as the means ± standard error. *** *p* < 0.001 and **** *p* < 0.0001 compared to untreated controls. AMK: *Atractylodes macrocephala* Koidz; CTRL: Control.

**Figure 6 nutrients-16-00965-f006:**
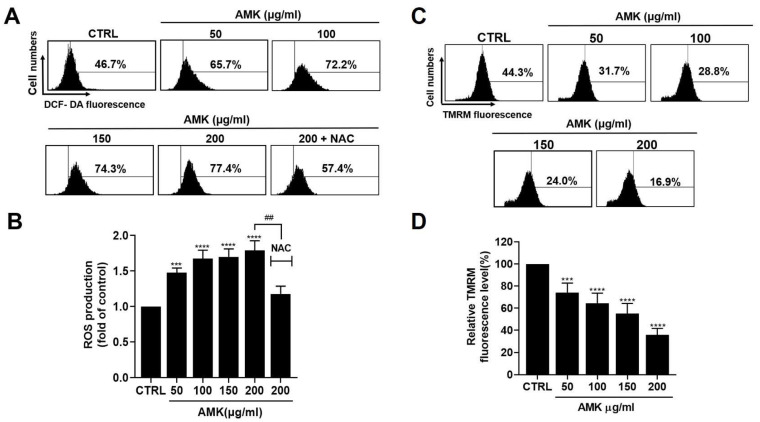
Effects of AMK extract on intracellular ROS levels and mitochondrial transmembrane potential in AGS cells. (**A**) Intracellular ROS levels were quantified using DCF-DA staining. (**B**) AMK extract increased the intracellular ROS levels. (**C**) The representative plots depict changes in mitochondrial membrane potential, which were evaluated through flow cytometry utilizing the mitochondrial membrane potential probe TMRM. (**D**) Quantitative results were obtained and normalized to the control group. Results are presented as the means ± standard error. *** *p* < 0.001 and **** *p* < 0.0001 compared to untreated controls. ## *p* < 0.01 compared to 200 μg/ml AMK. AMK: *Atractylodes macrocephala* Koidz; CTRL: Control.

**Figure 7 nutrients-16-00965-f007:**
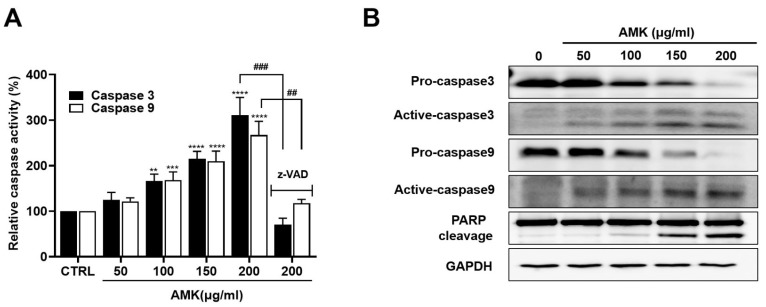
Effects of AMK extract on the activation of caspases and PARP in AGS cells. (**A**) Caspase-3 and Caspase-9 activities. Pre-incubation with z-VAD significantly inhibited AMK extract-induced caspase activity in AGS cells. (**B**) Western blot analysis assessed changes in Caspase-3, Caspase-9, and PARP cleavage activities in response to AMK extract treatment. Results are presented as the means ± standard error. ** *p* < 0.01, *** *p* < 0.001 and **** *p* < 0.0001 compared to untreated controls. ## *p* < 0.01, ### *p* < 0.001 compared to untreated with z-VAD. GAPDH served as the loading control. AMK: *Atractylodes macrocephala* Koidz; CTRL: Control.

**Figure 8 nutrients-16-00965-f008:**
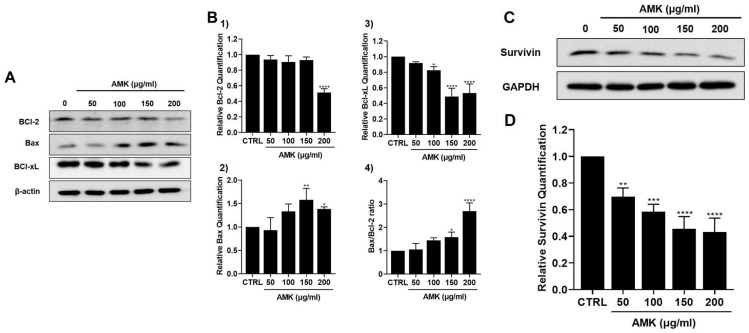
Effects of AMK extract on key apoptotic proteins in AGS cells. (**A**) AMK extract treatment led to a reduction in Bcl-2 levels, while concurrently inducing an increase in the levels of Bax, as evidenced by Western blot. (**B**) The quantified protein levels of (1) Bcl-2 and (2) Bax were subsequently normalized to the reference protein β-Actin. Additionally, AMK extract treatment reduced (3) Bcl-xL expression levels and (4) Bax/Bcl-2 ratio levels were gradually increased. (**C**) AMK extract reduced in survivin expression. (**D**) Survivin expression was normalized. Results are presented as the means ± standard error. * *p* < 0.05, ** *p* < 0.01, *** *p* < 0.001 and **** *p* < 0.0001 compared to untreated controls. AMK: *Atractylodes macrocephala* Koidz; CTRL: Control.

**Figure 9 nutrients-16-00965-f009:**
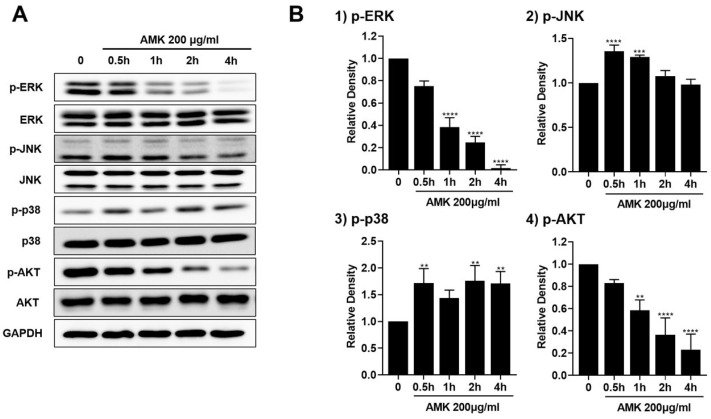
Effects of AMK extract on MAPK pathway in AGS cells. (**A**) Western blot was employed to assess the phosphorylation levels of key proteins in the MAPK pathway, including ERK, JNK, p38, and AKT. (**B**) The quantification of phosphorylated (1) ERK, (2) JNK, (3) p38, and (4) AKT was based on band densities relative to the density of GAPDH. Results are presented as the means ± standard error. ** *p* < 0.01, *** *p* < 0.001 and **** *p* < 0.0001 compared to untreated controls. AMK: *Atractylodes macrocephala* Koidz.

**Figure 10 nutrients-16-00965-f010:**
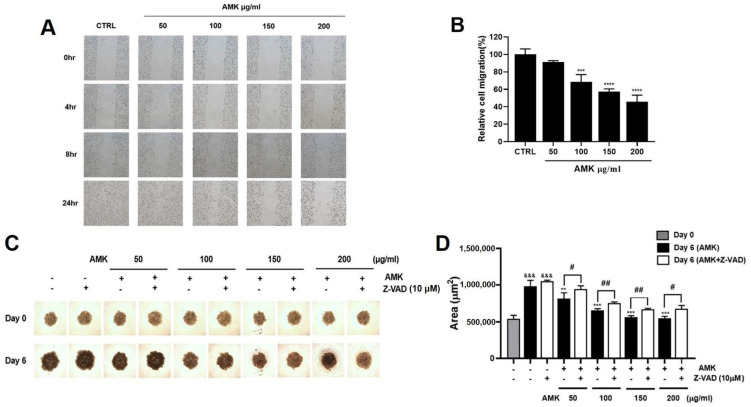
Effects of AMK extract on the migratory capacity and the growth within a 3D spheroid model in AGS cells. (**A**) Cells were imaged at 0, 4, 8, and 24 h after treatment with various AMK extract concentrations. (**B**) After 24 h treatment with AMK extract (50, 100, 150, and 200 μg/mL), the results showed a significant decrease in mobility. Spheroids were allowed to form over three days and then treated with AMK extract for six days using pre-treatment options including or excluding 10 μM z-VAD for 1 h before AMK extract treatment. (**C**) It shows a representative spheroid captured by phase contrast microscopy nine days after initial cell seeding (6 days after drug treatment). (**D**) The spheroid size was quantified using NIS-Elements BR software (Version 4.3). Results are presented as the means ± standard error. ** *p* < 0.01, *** *p* < 0.001 and **** *p* < 0.0001 compared to untreated controls. There is a significant size difference between spheroids on day 0 and day 6 of drug treatment (&&& *p* < 0.001) and spheroids without AMK extract treatment and spheroids treated with AMK extract have a significant difference. Differences between the Z-VAD treated group and the non-Z-VAD treated group are indicated as # *p* < 0.05, ## *p* < 0.01. AMK: *Atractylodes macrocephala* Koidz. CTRL: Control.

**Figure 11 nutrients-16-00965-f011:**
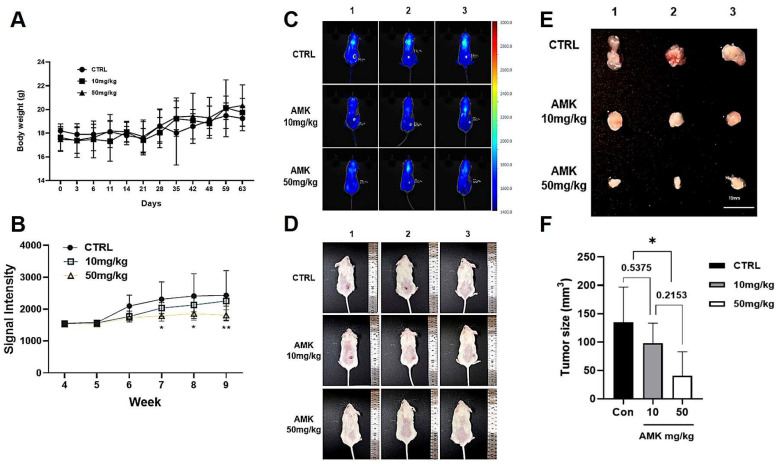
The body weight and tumor growth changes in xenograft mouse models. (**A**) There were no statistically significant differences in changes in body weight between different groups. (**B**) The signal intensity of the tumor was measured once a week for the AMK extract and vehicle-treated groups. (**C**,**D**) Images of tumors at the end of the experiment. (**E**,**F**) After 9 weeks of treatment, the tumor was excised and measured in size. Results are presented as the means ± standard error. * *p* < 0.05 and ** *p* < 0.01 compared to untreated controls. AMK: *Atractylodes macrocephala* Koidz; CTRL: Control.

**Figure 12 nutrients-16-00965-f012:**
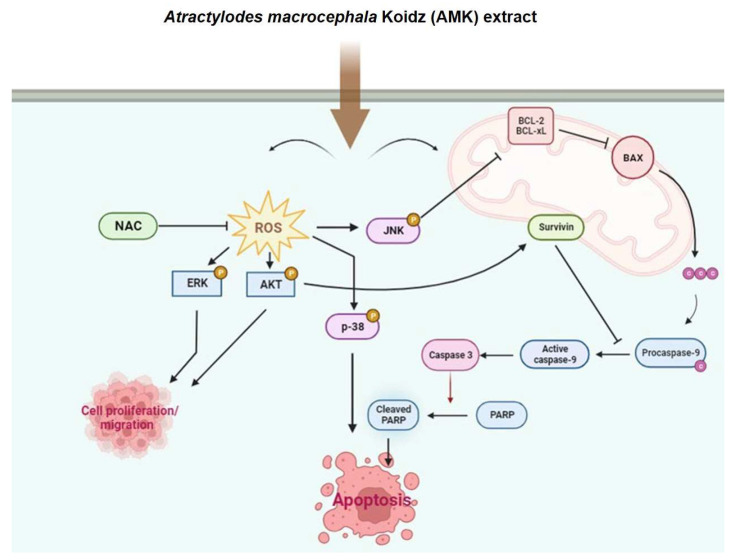
Systematic mechanisms for the apoptotic effect of AMK. AMK induces various ROS and mitochondria-dependent protein expressions, and these proteins induce apoptosis and regulate cell proliferation and migration of AGS cells by activating the ROS and intrinsic mitochondrial pathway. AMK: *Atractylodes macrocephala* Koidz; ROS, reactive oxygen species; BCl-2, B-cell lymphoma-2; Bax, Bcl-2-associated X protein; NAC, N-acetylcysteine; PARP, poly ADP-ribose polymerase; BCL-xL, B-cell lymphoma-extra large; ERK, extracellular signal-regulated kinase; JNK, c-Jun N-terminal kinase; AKT, Serine/Threonine protein kinase.

**Table 1 nutrients-16-00965-t001:** Compounds and targets related to GI cancer.

Molecule Name	Gene Name	Disease Name
(+/−)-Isoborneol	PTGS2	Cancer, unspecific
		Carcinoma in situ, unspecified
		Colorectal cancer
		Oropharyngeal squamous cell
		Carcinoma
(1R)-2-methyl-1-phenylprop-2-en-1-ol	DPP4	Malignancies
(1S,2R,4R)-Neoiso-dihydrocarveol	REN	Cancer, unspecific
(3S)-3-[(1R)-1,5-dimethylhex-4-enyl]-6-methylenecyclohexene	PTGS2	Cancer, unspecific
		Carcinoma in situ, unspecified
		Colorectal cancer
		Oropharyngeal squamous cell
		Carcinoma
(5E,9Z)-3,6,10-trimethyl-4,7,8,11-tetrahydrocyclodeca[b]furan	NOS3	Colon cancer
	PTGS2	Cancer, unspecific
		Carcinoma in situ, unspecified
		Colorectal cancer
		Oropharyngeal squamous cell
		Carcinoma
14-acetyl-12-senecioyl-2E,8Z,10E-atractylentriol	PTGS2	Cancer, unspecific
		Carcinoma in situ, unspecified
		Colorectal cancer
		Oropharyngeal squamous cell
		Carcinoma
3β-acetoxyatractylone	DPP4	Malignancies
	NOS3	Colon cancer
	PTGS2	Cancer, unspecific
		Carcinoma in situ, unspecified
		Colorectal cancer
		Oropharyngeal squamous cell
		Carcinoma
8β-ethoxy atractylenolide III	PTGS2	Cancer, unspecific
		Carcinoma in situ, unspecified
		Colorectal cancer
		Oropharyngeal squamous cell
		Carcinoma
Akridin	PTGS2	Cancer, unspecific
		Carcinoma in situ, unspecified
		Colorectal cancer
		Oropharyngeal squamous cell
		Carcinoma
alpha-Curcumene	PTGS2	Cancer, unspecific
		Carcinoma in situ, unspecified
		Colorectal cancer
		Oropharyngeal squamous cell
		Carcinoma
alpha-humulene	TNF	* gastric cancer
		Solid Tumor
	PTGS2	Cancer, unspecific
		Carcinoma in situ, unspecified
		Colorectal cancer
		Oropharyngeal squamous cell
		Carcinoma
	REN	Cancer, unspecific
ASI	ALOX5	Gastrointestinal Cancers
		Pancreatic Cancer
		Gastrointestinal Cancers
		Pancreatic Cancer
	CDC25B	Cancer, unspecific
	PTPN1	Cancer, unspecific
atractylenolide i	IL6	* gastric cancer
	TNF	* gastric cancer
		Solid Tumor
	VEGFA	Colorectal Neoplasms
atractylenolide iii	TNF	* gastric cancer
		Solid Tumor
atractylone	NOS3	Colon cancer
beta-caryophyllene	IL6	* gastric cancer
	PTGS2	Cancer, unspecific
		Carcinoma in situ, unspecified
		Colorectal cancer
		Oropharyngeal squamous cell
		Carcinoma
beta-Humulene	PTGS2	Cancer, unspecific
		Carcinoma in situ, unspecified
		Colorectal cancer
		Oropharyngeal squamous cell
		Carcinoma
beta-Selinene	PTGS2	Cancer, unspecific
		Carcinoma in situ, unspecified
		Colorectal cancer
		Oropharyngeal squamous cell
		Carcinoma
D-Camphene	PTGS2	Cancer, unspecific
		Carcinoma in situ, unspecified
		Colorectal cancer
		Oropharyngeal squamous cell
		Carcinoma
	TOP2A	Cancer, unspecific
D-Serin	SRC	* gastric cancer
		Cancer, unspecific
DTY	DPP4	Malignancies
	NOS3	Colon cancer
	PTGS2	Cancer, unspecific
		Carcinoma in situ, unspecified
		Colorectal cancer
		Oropharyngeal squamous cell
		Carcinoma
	PTPN1	Cancer, unspecific
GLY	CTNNB1	* gastric cancer
		Colorectal cancer
	ALDH1A1	Neoplasms
	CDC25B	Cancer, unspecific
	LTA4H	Oesophageal cancer
		Solid tumors
	MMP8	Tumors
	PTGS2	Cancer, unspecific
		Carcinoma in situ, unspecified
		Colorectal cancer
		Oropharyngeal squamous cell
		Carcinoma
	PTPN1	Cancer, unspecific
	REN	Cancer, unspecific
	RRM1	Pancreatic Neoplasms
	SHMT2	Cancer, unspecific
	SRC	Cancer, unspecific
	TXNRD1	Cancer, unspecific
		Malignancies
Gulutamine	LTA4H	Oesophageal cancer
		Solid tumors
	PTGS2	Cancer, unspecific
		Carcinoma in situ, unspecified
		Colorectal cancer
		Oropharyngeal squamous cell
		Carcinoma
	PTPN1	Cancer, unspecific
Hemo-sol	PTGS2	Cancer, unspecific
		Carcinoma in situ, unspecified
		Colorectal cancer
		Oropharyngeal squamous cell
		Carcinoma
Istidina	PTGS2	Cancer, unspecific
		Carcinoma in situ, unspecified
		Colorectal cancer
		Oropharyngeal squamous cell
		Carcinoma
juniper camphor	PTGS2	Cancer, unspecific
		Carcinoma in situ, unspecified
		Colorectal cancer
		Oropharyngeal squamous cell
		Carcinoma
L-Arginin	PTGS2	Cancer, unspecific
		Carcinoma in situ, unspecified
		Colorectal cancer
		Oropharyngeal squamous cell
		Carcinoma
L-Ile	REN	Cancer, unspecific
LPG	SRC	* gastric cancer
		Cancer, unspecific
	CDC25B	Cancer, unspecific
	MMP8	Tumors
	RRM1	Pancreatic Neoplasms
L-Valin	PTGS2	Cancer, unspecific
		Carcinoma in situ, unspecified
		Colorectal cancer
		Oropharyngeal squamous cell
		Carcinoma
	PTPN1	Cancer, unspecific
palmitic acid	BCL2	* gastric cancer
	PTEN	* gastric cancer
	TNF	* gastric cancer
		Solid Tumor
	PTGS2	Cancer, unspecific
		Carcinoma in situ, unspecified
		Colorectal cancer
		Oropharyngeal squamous cell
		Carcinoma
PHA	DPP4	Malignancies
	NOS3	Colon cancer
	PTGS2	Cancer, unspecific
		Carcinoma in situ, unspecified
		Colorectal cancer
		Oropharyngeal squamous cell
		Carcinoma
Prolinum	PTGS2	Cancer, unspecific
		Carcinoma in situ, unspecified
		Colorectal cancer
		Oropharyngeal squamous cell
		Carcinoma
Scopoletol	CA1	Pancreatic Cancer
	LTA4H	Oesophageal cancer
		Solid tumors
	NQO2	Cancer, unspecific
		Tumors
	PTGS2	Cancer, unspecific
		Carcinoma in situ, unspecified
		Colorectal cancer
		Oropharyngeal squamous cell
		Carcinoma
selina-4(14),7(11)-dien-8-one	PTGS2	Cancer, unspecific
		Carcinoma in situ, unspecified
		Colorectal cancer
		Oropharyngeal squamous cell
		Carcinoma
α-Longipinene	PTGS2	Cancer, unspecific
		Carcinoma in situ, unspecified
		Colorectal cancer
		Oropharyngeal squamous cell
		Carcinoma
γ-elemene	PTGS2	Cancer, unspecific
		Carcinoma in situ, unspecified
		Colorectal cancer
		Oropharyngeal squamous cell
		Carcinoma

* After investigating the relationship between *Atractylodes Macrocephala* Koidz. and gastric cancer using Cytoscape, gastric-cancer-related genes were added to this table.

**Table 2 nutrients-16-00965-t002:** One hundred gastric-cancer-related genes.

Gene Name	Protein Name
*AFP*	Alpha-1-fetoprotein
*AKR7A3*	Aldo-keto reductase family 7, member A3 (aflatoxin aldehyde reductase)
*AKT1*	V-akt murine thymoma viral oncogene homolog 1
*ALB*	Serum albumin
*ANXA5*	Placental anticoagulant protein 4
*ARID1A*	SWI/SNF-related, matrix-associated, actin-dependent regulator of chromatin subfamily F member 1
*ATP12A*	ATPase, H+/K+ transporting, nongastric, alpha polypeptide
*ATP4A*	ATPase, H+/K+ exchanging, alpha polypeptide
*ATP4B*	ATPase, H+/K+ exchanging, beta polypeptide
*BCL2*	Apoptosis regulator Bcl-2
*BRCA1*	Breast cancer type 1 susceptibility protein
*BRCA2*	Breast cancer type 2 susceptibility protein
*C3orf36*	Chromosome 3 open reading frame 36
*CAGE1*	Cancer-associated gene 1 protein
*CASP3*	Caspase 3, apoptosis-related cysteine peptidase
*CASP9*	Caspase 9, apoptosis-related cysteine peptidase
*CCND1*	B-cell lymphoma 1 protein
*CCNE1*	G1/S-specific cyclin-E1
*CD274*	Programmed cell death 1 ligand 1
*CD4*	T-cell surface antigen T4/Leu-3
*CD44*	GP90 lymphocyte homing/adhesion receptor
*CD8A*	T-lymphocyte differentiation antigen T8/Leu-2
*CDH1*	Cadherin 1, type 1, E-cadherin (epithelial)
*CDH2*	Cadherin 2, type 1, N-cadherin (neuronal)
*CDX2*	Caudal-type homeobox protein 2
*CEACAM5*	Carcinoembryonic antigen-related cell adhesion molecule 5
*CGB*	Chorionic gonadotropin, beta polypeptide
*CLDN18*	Claudin 18
*COL5A2*	Collagen alpha-2(V) chain
*CT83*	Kita-kyushu lung cancer antigen 1
*CTLA4*	Cytotoxic T-lymphocyte-associated antigen 4
*CTNNA1*	Catenin (cadherin-associated protein), alpha 1, 102kDa
*CTNNB1*	Catenin (cadherin-associated protein), beta 1, 88kDa
*DIRC1*	Disrupted in renal carcinoma 1
*EGFR*	Receptor tyrosine-protein kinase erbB-1
*ENSP00000464218*	Zinc finger protein 286A
*ERBB2*	V-erb-b2 avian erythroblastic leukemia viral oncogene homolog 2
*ERBB3*	V-erb-b2 avian erythroblastic leukemia viral oncogene homolog 3
*EZH2*	Enhancer of zeste 2 polycomb repressive complex 2 subunit
*FGFR2*	Fibroblast growth factor receptor 2
*FNDC1*	Fibronectin type III domain-containing protein 1
*FNDC7*	Fibronectin type III domain-containing 7
*FNDC8*	Fibronectin type III domain containing 8
*FTO*	Alpha-ketoglutarate-dependent dioxygenase FTO
*GAST*	Gastrin
*GLT8D2*	Glycosyltransferase 8 domain containing 2
*GPR176*	G protein-coupled receptor 176
*GPX4*	Phospholipid hydroperoxide glutathione peroxidase, mitochondrial
*HAVCR2*	T-cell immunoglobulin and mucin domain-containing protein 3
*HIF1A*	Hypoxia inducible factor 1, alpha subunit (basic helix-loop-helix transcription factor)
*HRAS*	Harvey rat sarcoma viral oncogene homolog
*IL6*	B-cell stimulatory factor 2
*KDR*	Kinase insert domain receptor (a type III receptor tyrosine kinase)
*KIAA1524*	Cancerous inhibitor of PP2A
*KIT*	V-kit Hardy-Zuckerman 4 feline sarcoma viral oncogene homolog
*KRAS*	Kirsten rat sarcoma viral oncogene homolog
*KRT7*	Keratin, type II cytoskeletal 7
*LAG3*	Lymphocyte activation gene 3 protein
*LGR5*	Leucine-rich repeat containing G protein-coupled receptor 5
*LIPT1*	Lipoyltransferase 1, mitochondrial
*LRP1B*	Low-density lipoprotein receptor-related protein-deleted in tumor
*LRRD1*	Leucine rich repeats and death domain containing 1
*MAGEA11*	Melanoma-associated antigen 11
*MAP6*	Microtubule-associated protein 6
*METTL3*	N6-adenosine-methyltransferase catalytic subunit
*MLH1*	DNA mismatch repair protein Mlh1
*MMP2*	Matrix metallopeptidase 2 (gelatinase A, 72 kDa gelatinase, 72 kDa type IV collagenase)
*MMP9*	Matrix metallopeptidase 9 (gelatinase B, 92 kDa gelatinase, 92 kDa type IV collagenase)
*MSH2*	DNA mismatch repair protein Msh2
*MSH6*	DNA mismatch repair protein Msh6
*MTUS1*	Angiotensin-II type 2 receptor-interacting protein
*MTUS2*	Microtubule associated tumor suppressor candidate 2
*MUC5AC*	Mucin 5AC, oligomeric mucus/gel-forming
*MUC6*	Mucin 6, oligomeric mucus/gel-forming
*MYC*	V-myc avian myelocytomatosis viral oncogene homolog
*NOTCH1*	Translocation-associated notch protein TAN-1
*OLFML2B*	Olfactomedin-like protein 2B
*PDCD1*	Programmed cell death protein 1
*PIK3CA*	Phosphatidylinositol 4,5-bisphosphate 3-kinase 110 kDa catalytic subunit alpha
*PLOD2*	Procollagen-lysine, 2-oxoglutarate 5-dioxygenase 2
*PMS2*	PMS2 postmeiotic segregation increased 2 (*S. cerevisiae*)
*PSCA*	Prostate stem cell antigen
*PTEN*	Mutated in multiple advanced cancers 1
*RNF180*	RING-type E3 ubiquitin transferase RNF180
*RUNDC3A*	RUN domain-containing protein 3A
*S100A8*	Migration inhibitory factor-related protein 8
*SALL4*	Spalt-like transcription factor 4
*SLC7A11*	Solute carrier family 7 (anionic amino acid transporter light chain, xc-system), member 11
*SNAI1*	Snail family zinc finger 1
*SOX2*	SRY (sex determining region Y)-box 2
*SRC*	SRC proto-oncogene, non-receptor tyrosine kinase
*STAT3*	Signal transducer and activator of transcription 3 (acute-phase response factor)
*TFF2*	Spasmolytic polypeptide
*TNF*	Tumor necrosis factor ligand superfamily member 2
*TP53*	Cellular tumor antigen p53
*TRIM49C*	Tripartite motif-containing protein 49-like protein 2
*VSIG1*	V-set and immunoglobulin domain containing 1
*YAP1*	Yes-associated protein YAP65 homolog
*YTHDF1*	Dermatomyositis associated with cancer putative autoantigen 1
*ZNF705A*	Zinc finger protein 705A

## Data Availability

The original data are available upon reasonable request to the corresponding author. The data are not publicly available due to privacy and ethical restrictions.

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
