# Peer review of "Exploring the Therapeutic Effects of Atractylodes macrocephala Koidz against Human Gastric Cancer"

_nutrients, 2024, doi:10.3390/nu16070965_

Round 1

Reviewer 1 Report

Comments and Suggestions for Authors

The authors describe very well the antitumour effect of AMK extract on gastric cancer using in vitro and in vivo approaches. However, the manuscript is in need of some minor revisions as follows: 

INTRODUCTIONS. LINE 42: Please change stomac cancer with gastric cancer. 

MATERIALS AND METHODS: SECTION 2.2.10. LINE 152: Please explain in this section what is z-VAD. 

RESULTS: Use GI or GC to indicate gastric tumour. 

SECTION 3.2.3. LINES 257-258: please add a reference

SECTION 3.2.7. LINES 321-323:please add a reference

Author Response

The authors describe very well the antitumour effect of AMK extract on gastric cancer using in vitro and in vivo approaches. However, the manuscript is in need of some minor revisions as follows: 

INTRODUCTIONS. LINE 42: Please change stomac cancer with gastric cancer. 

Responses) We changed stomach cancer with gastric cancer.

MATERIALS AND METHODS:

SECTION 2.2.10. LINE 152: Please explain in this section what is z-VAD. 

Responses) We explained the z-VAD. Z-VAD is pan-caspase inhibitor.

RESULTS: Use GI or GC to indicate gastric tumour. 

Responses) GI cancer refers to cancer that appears in the entire gastrointestinal tract, and gastric cancer refers to cancer that appears only in the stomach. Therefore, it is supplemented in the text. (In lines 200. Gastric cancer among GI cancer).

SECTION 3.2.3. LINES 257-258: please add a reference

Responses) We added a reference.

11.Vermeulen, K.; Berneman, Z.N.; Van Bockstaele, D.R. Cell cycle and apoptosis. Cell. Prolif. 2003, 36, 165-175.

SECTION 3.2.7. LINES 321-323:please add a reference

Responses) We added a reference.

17.Leibowitz, B.; Yu, J. Mitochondrial signaling in cell death via the Bcl-2 family. Cancer Biol. Ther. 2010, 9, 417-422.

Reviewer 2 Report

Comments and Suggestions for Authors

The author has done a lot of research work, but there are the following problems: 1. Abstract conclusions: AMK extract exert its anti-cancer effect by promoting the apoptosis of gastric cancer cells, but from the results of the manuscript, AMK extract  also exert anti-cancer by promoting cell cycle arrest and inhibiting the proliferation of cancer cells, therefore, the conclusion in the abstract is inappropriate.

2. The authors did a network pharmacological study on the association of AMK active components with gastric cancer, predicting some potential target genes, but did not do experiments to confirm this, which needs to add new experiments and confirm is right or not.

3. The authors used bioinformatics to analyze the AMK activity component associated with gastric cancer, but did not discuss them in the discussion part.

4. Caspase-3 is referred to genes in lowercasecaspase-3and proteins in capitalCaspase-3.. as well as beta-actin and beta-Actin

5. In vivo, in vitro, manuscript did not use italics in some place, such as abstract part. 

6. Language problem and errors, such as 3.0 × 105 cells/mL

Comments on the Quality of English Language

1. Caspase-3 is referred to genes in lowercasecaspase-3” and proteins in capitalCaspase-3.. as well as beta-actin and beta-Actin

2. In vivo, in vitro, manuscript did not use italics in some place, such as abstract part. 

3. Language problem and errors, such as 3.0 × 105 cells/mL

Author Response

The author has done a lot of research work, but there are the following problems:

1.Abstract conclusions: AMK extract exert its anti-cancer effect by promoting the apoptosis of gastric cancer cells, but from the results of the manuscript, AMK extract also exert anti-cancer by promoting cell cycle arrest and inhibiting the proliferation of cancer cells, therefore, the conclusion in the abstract is inappropriate.

Responses) We changed the abstract conclusions.

2.The authors did a network pharmacological study on the association of AMK active components with gastric cancer, predicting some potential target genes, but did not do experiments to confirm this, which needs to add new experiments and confirm is right or not.

Responses) This is a very critical comment and we understand the value of experimental evidences of the connection between AMK and the six genes identified by the network analysis. In our study, TNF, IL6, BCL2, PTEN, CTNNB1, and SRC were presented as common target genes between gastric cancer and AMK as you pointed out (Figure 2). We believe it is difficult to confirm the relevance of these genes during the revision of this manuscript. However, in the previous studies, we have validated the relationship between these six genes and gastric cancer. TNF has been known to enhance inflammatory responses in gastric cancer [1]. Among TNFs, TNF-alpha is mainly involved in the early stages of gastric cancer [1]. Therefore, TNF-alpha is the most studied marker in gastric cancer and many studies are being conducted on the gastric cancer development and the relationship with other gastric cancer biomarkers [2]. IL-6 is one of the most important cytokines in the tumor microenvironment and many studies have proved that IL-6 is involved in cancer cell growth and metastasis [3]. In gastric cancer patient blood, IL-6 concentration is higher in stage 4 than in stages 2 and 3, so IL-6 may be related to the malignancy of gastric cancer such as metastasis and cancer severity [4]. BCL2 inhibits the mitochondrial pathway of apoptosis and plays an important role in the progression of gastric cancer, which is tightly related to the severity of gastric cancer [5]. It has been known that BCL2 expression level is increased, as gastric cancer progresses [6]. PTEN regulates various cellular processes, such as the induction of apoptosis [7]. When PTEN activity decreases, it has been known that it is associated with various cancers, and gastric cancer is also known to be associated with PTEN inactivation [8]. CTNNB1 genes are present in gastric cancer and are involved in the signaling mechanisms of gastric cancer along with various molecules [9]. Various signaling such as Wnt, Notch, and Ephrin have been needed to reveal the CTNNB1 signaling in cancer [9]. SRC has been involved in the growth and metastasis of cancer cells and is also an important target for gastric cancer treatment [10]. c-SRC was the first oncogene discovered and has increased expression in the gastric cancer [11,12]. As such, there are many previous reports that the six genes identified by the network analysis are related to gastric cancer, and further clarification is needed in the future to reveal the underlying mechanisms of AMK.

<References>

1.Roșu, M.C.; Mihnea, P.D.; Ardelean, A.; Moldovan, S.D.; Popețiu, R.O.; Totolici, B.D. Clinical significance of tumor necrosis factor-alpha and carcinoembryonic antigen in gastric cancer. J. Med. Life 2022, 15, 4-6.

2.Oshima, H.; Ishikawa, T.; Yoshida, G.J.; Naoi, K.; Maeda, Y.; Naka, K.; Ju, X.; Yamada, Y.; Minamoto, T.; Mukaida, N.; et al. TNF-α/TNFR1 signaling promotes gastric tumorigenesis through induction of Noxo1 and Gna14 in tumor cells. Oncogene 2014, 33, 3820-3829.

3.Wang X, Li J, Liu W, Zhang X, Xue L. The diagnostic value of interleukin 6 as a biomarker for gastric cancer: A meta-analysis and systematic review. Medicine (Baltimore) 2021, 100, e27945.

4.Ito, R.; Yasui, W.; Kuniyasu, H.; Yokozaki, H.; Tahara, E. Expression of interleukin-6 and its effect on the cell growth of gastric carcinoma cell lines. Jpn. J. Cancer Res. 1997, 88, 953–958.

5.Gryko, M.; Pryczynicz, A.; Zareba, K.; Kędra, B.; Kemona, A.; Guzińska-Ustymowicz, K. The expression of Bcl-2 and BID in gastric cancer cells. J. Immunol. Res. 2014, 2014, 953203.

6.Cory, S.; Adams, J.M, The Bcl2 family: regulators of the cellular life-or-death switch. Nat. Rev. Cancer 2002, 2, 647–656.

7.Xu, W.T.; Yang, Z.; Lu, N.H. Roles of PTEN (Phosphatase and Tensin Homolog) in gastric cancer development and progression. Asian Pac. J. Cancer Prev. 2014, 15, 17–24.

8.Kim, B.; Kang, S.Y.; Kim, D.; Heo, Y.J.; Kim, K.M. PTEN Protein Loss and Loss-of-Function Mutations in Gastric Cancers: The Relationship with Microsatellite Instability, EBV, HER2, and PD-L1 Expression. Cancers (Basel) 2020, 12, 1724.

9.Tanabe, S.; Kawabata, T.; Aoyagi, K.; Yokozaki, H.; Sasaki, H. Gene expression and pathway analysis of CTNNB1 in cancer and stem cells. World J. Stem Cells 20168, 384-395.

10.DE Fátima Ferreira Borges DA Costa, J.; DE Castro Sant' Anna, C.; Muniz, J.A.P.C.; DA Rocha, C.A.M.; Lamarão, L.M.; DE Fátima Aquino Moreira Nunes, C.; DE Assumpção, P.P.; Burbano, R.R. Deregulation of the SRC Family Tyrosine Kinases in Gastric Carcinogenesis in Non-human Primates. Anticancer Res. 2018, 38, 6317-6320.

11.Yeatman, T.J. A renaissance for SRC. Nat. Rev. Cancer 2004, 4, 470-480.

12.Nam, H.J.; Im, S.A.; Oh, D.Y.; Elvin, P.; Kim, H.P.; Yoon, Y.K.; Min, A.; Song, S.H.; Han, S.W.; Kim, T.Y.; et al. Antitumor activity of saracatinib (AZD0530), a c-Src/Abl kinase inhibitor, alone or in combination with chemotherapeutic agents in gastric cancer. Mol. Cancer Ther. 2013, 12, 16-26.

3.The authors used bioinformatics to analyze the AMK activity component associated with gastric cancer, but did not discuss them in the discussion part.

Responses) We added this point in the discussion part.

As a result of network pharmacological analysis, 55 compounds containing 18 active compounds in AMK were confirmed. 50 of the 55 compounds had target information, and 260 targeted genes were collected (Supplementary Materials Table S1 and Figure S1). Among them, GLY was linked to the most significant number of target genes.

In our study, TNF, IL6, BCL2, PTEN, CTNNB1, and SRC were presented as common target genes between gastric cancer and AMK (Figure 2). We believe it is difficult to confirm the relevance of these genes during the revision of this manuscript. However, in the previous studies, we have validated the relationship between these six genes and gastric cancer. TNF has been known to enhance inflammatory responses in gastric cancer [1]. Among TNFs, TNF-alpha is mainly involved in the early stages of gastric cancer [1]. Therefore, TNF-alpha is the most studied marker in gastric cancer and many studies are being conducted on the gastric cancer development and the relationship with other gastric cancer biomarkers [2]. IL-6 is one of the most important cytokines in the tumor microenvironment and many studies have proved that IL-6 is involved in cancer cell growth and metastasis [3]. In gastric cancer patient blood, IL-6 concentration is higher in stage 4 than in stages 2 and 3, so IL-6 may be related to the malignancy of gastric cancer such as metastasis and cancer severity [4]. BCL2 inhibits the mitochondrial pathway of apoptosis and plays an important role in the progression of gastric cancer, which is tightly related to the severity of gastric cancer [5]. It has been known that BCL2 expression level is increased, as gastric cancer progresses [6]. PTEN regulates various cellular processes, such as the induction of apoptosis [7]. When PTEN activity decreases, it has been known that it is associated with various cancers, and gastric cancer is also known to be associated with PTEN inactivation [8]. CTNNB1 genes are present in gastric cancer and are involved in the signaling mechanisms of gastric cancer along with various molecules [9]. Various signaling such as Wnt, Notch, and Ephrin have been needed to reveal the CTNNB1 signaling in cancer [9]. SRC has been involved in the growth and metastasis of cancer cells and is also an important target for gastric cancer treatment [10]. c-SRC was the first oncogene discovered and has increased expression in the gastric cancer [11,12]. As such, there are many previous reports that the six genes identified by the network analysis are related to gastric cancer, and further clarification is needed in the future to reveal the underlying mechanisms of AMK.

As shown in Figure 3, AMK compounds targeting gastric cancer related genes, palmitic acid, atractylenolide I, GLY, beta-caryophyllene, D-Serine, LPG, alpha-humulene, and atractylenolide III were identified. In addition, palmitic acid was observed to target PTEN and BLC2. Atractylenolide I targeted TNF and IL6 and GLY targeted CTNNB1. Beta-caryophyllene targeted IL6, D-Serine and LPG targeted SRC, and alpha-humulene and atractylenolide III targeted TNF. These results show the characteristics of multiple compounds-multiple targets of traditional medicines, and with the synergistic effect of multiple compounds-multiple targets of AMK, we could predict the therapeutic effects of AMK on gastric cancer.

<References>

1.Roșu, M.C.; Mihnea, P.D.; Ardelean, A.; Moldovan, S.D.; Popețiu, R.O.; Totolici, B.D. Clinical significance of tumor necrosis factor-alpha and carcinoembryonic antigen in gastric cancer. J. Med. Life 2022, 15, 4-6.

2.Oshima, H.; Ishikawa, T.; Yoshida, G.J.; Naoi, K.; Maeda, Y.; Naka, K.; Ju, X.; Yamada, Y.; Minamoto, T.; Mukaida, N.; et al. TNF-α/TNFR1 signaling promotes gastric tumorigenesis through induction of Noxo1 and Gna14 in tumor cells. Oncogene 2014, 33, 3820-3829.

3.Wang X, Li J, Liu W, Zhang X, Xue L. The diagnostic value of interleukin 6 as a biomarker for gastric cancer: A meta-analysis and systematic review. Medicine (Baltimore) 2021, 100, e27945.

4.Ito, R.; Yasui, W.; Kuniyasu, H.; Yokozaki, H.; Tahara, E. Expression of interleukin-6 and its effect on the cell growth of gastric carcinoma cell lines. Jpn. J. Cancer Res. 1997, 88, 953–958.

5.Gryko, M.; Pryczynicz, A.; Zareba, K.; Kędra, B.; Kemona, A.; Guzińska-Ustymowicz, K. The expression of Bcl-2 and BID in gastric cancer cells. J. Immunol. Res. 2014, 2014, 953203.

6.Cory, S.; Adams, J.M, The Bcl2 family: regulators of the cellular life-or-death switch. Nat. Rev. Cancer 2002, 2, 647–656.

7.Xu, W.T.; Yang, Z.; Lu, N.H. Roles of PTEN (Phosphatase and Tensin Homolog) in gastric cancer development and progression. Asian Pac. J. Cancer Prev. 2014, 15, 17–24.

8.Kim, B.; Kang, S.Y.; Kim, D.; Heo, Y.J.; Kim, K.M. PTEN Protein Loss and Loss-of-Function Mutations in Gastric Cancers: The Relationship with Microsatellite Instability, EBV, HER2, and PD-L1 Expression. Cancers (Basel) 2020, 12, 1724.

9.Tanabe, S.; Kawabata, T.; Aoyagi, K.; Yokozaki, H.; Sasaki, H. Gene expression and pathway analysis of CTNNB1 in cancer and stem cells. World J. Stem Cells 20168, 384-395.

10.DE Fátima Ferreira Borges DA Costa, J.; DE Castro Sant' Anna, C.; Muniz, J.A.P.C.; DA Rocha, C.A.M.; Lamarão, L.M.; DE Fátima Aquino Moreira Nunes, C.; DE Assumpção, P.P.; Burbano, R.R. Deregulation of the SRC Family Tyrosine Kinases in Gastric Carcinogenesis in Non-human Primates. Anticancer Res. 2018, 38, 6317-6320.

11.Yeatman, T.J. A renaissance for SRC. Nat. Rev. Cancer 2004, 4, 470-480.

12.Nam, H.J.; Im, S.A.; Oh, D.Y.; Elvin, P.; Kim, H.P.; Yoon, Y.K.; Min, A.; Song, S.H.; Han, S.W.; Kim, T.Y.; et al. Antitumor activity of saracatinib (AZD0530), a c-Src/Abl kinase inhibitor, alone or in combination with chemotherapeutic agents in gastric cancer. Mol. Cancer Ther. 2013, 12, 16-26.

4.Caspase-3 is referred to genes in lowercase“caspase-3” and proteins in capital“Caspase-3.”. as well as beta-actin and beta-Actin.

Responses) We checked again and corrected it.

5.In vivo, in vitro, manuscript did not use italics in some place, such as abstract part. 

Responses) We checked again and corrected it.

6.Language problem and errors, such as 3.0 × 105 cells/mL

Responses) We checked again and corrected it.

Comments on the Quality of English Language

1.Caspase-3 is referred to genes in lowercase“caspase-3” and proteins in capital“Caspase-3.”. as well as beta-actin and beta-Actin

Responses) We checked again and corrected it.

2.In vivo, in vitro, manuscript did not use italics in some place, such as abstract part. 

Responses) We checked again and corrected it.

3.Language problem and errors, such as 3.0 × 105 cells/mL

Responses) We checked again and corrected it.

Round 2

Reviewer 2 Report

Comments and Suggestions for Authors

OK